# Automated Path Searching Reveals the Mechanism of Hydrolysis Enhancement by T4 Lysozyme Mutants

**DOI:** 10.3390/ijms232314628

**Published:** 2022-11-23

**Authors:** Kun Xi, Lizhe Zhu

**Affiliations:** Warshel Institute for Computational Biology, School of Life and Health Sciences, School of Medicine, The Chinese University of Hong Kong, Shenzhen 518172, China

**Keywords:** bacteriophage T4 lysozyme, path searching, rational protein engineering

## Abstract

Bacteriophage T4 lysozyme (T4L) is a glycosidase that is widely applied as a natural antimicrobial agent in the food industry. Due to its wide applications and small size, T4L has been regarded as a model system for understanding protein dynamics and for large-scale protein engineering. Through structural insights from the single conformation of T4L, a series of mutations (L99A,G113A,R119P) have been introduced, which have successfully raised the fractional population of its only hydrolysis-competent excited state to 96%. However, the actual impact of these substitutions on its dynamics remains unclear, largely due to the lack of highly efficient sampling algorithms. Here, using our recently developed travelling-salesman-based automated path searching (TAPS), we located the minimum-free-energy path (MFEP) for the transition of three T4L mutants from their ground states to their excited states. All three mutants share a three-step transition: the flipping of F114, the rearrangement of α0/α1 helices, and final refinement. Remarkably, the MFEP revealed that the effects of the mutations are drastically beyond the expectations of their original design: (a) the G113A substitution not only enhances helicity but also fills the hydrophobic Cavity I and reduces the free energy barrier for flipping F114; (b) R119P barely changes the stability of the ground state but stabilizes the excited state through rarely reported polar contacts S117_OG_:N132_ND2_, E11_OE1_:R145_NH1_, and E11_OE2_:Q105_NE2_; (c) the residue W138 flips into Cavity I and further stabilizes the excited state for the triple mutant L99A,G113A,R119P. These novel insights that were unexpected in the original mutant design indicated the necessity of incorporating path searching into the workflow of rational protein engineering.

## 1. Introduction

Bacteriophage T4 lysozyme (T4L) is a glycosidase that can attack and hydrolyze the peptidoglycans in the cell walls of bacteria [1,2,3,4]. T4L has been widely applied as a natural antimicrobial agent in the food industry [5] and the fusion of proteins to facilitate the crystallization of G-protein-coupled receptors [6,7,8,9], sterol-binding proteins [10,11] and type II phosphatidylinositol 4-kinases [12].

As a 164-residue polypeptide, T4L folds into a helix-rich bilobed structure [13]. It contains two important hydrophobic cavities that directly influence its activation: hydrophobic Cavity I, which is composed of M102/M106/L133/W138 (cyan in Figure 1A), and hydrophobic Cavity II, which is composed of L84/V87/A99/V103 (green in Figure 1A) [14,15]. These two hydrophobic cavities are separated by two alpha helices: α0 and α1 (see Figure 1A).

Due to its wide applications and relatively small size, T4L has been regarded as a representative model system for understanding the relationships among a protein’s structure, dynamics, and functions [16,17,18,19], which has triggered a large body of protein-engineering studies [4,15,20,21,22,23,24]. Most of such engineering attempts aimed to find a T4L variant with a sufficiently populated excited state (E state, Figure 1A). As the bare conformational state with hydrolysis capability, the E state of wild-type T4L is only transiently formed and marginally populated and, therefore, unable to be individually characterized by most biophysical tools [14,15].

To enhance the efficiency of T4L hydrolysis, three variants were designed. The L99A mutant (T4L-L99A), first made to strengthen the hydrophobicity of T4L [20], significantly broadened the peaks of solution NMR relative to the wide-type (WT) and raised the fractional population of the E state to 3% (97% for its ground state, or G state) [15]. Subsequently, as Rosetta modeling predicted that substituting G113 by alanine might enhances the helicity of α1, the G113A mutation was introduced to T4L-L99A, resulting in an additional increase in the E state population to 34% [15]. In the course of destabilizing the G state, the R119P substitution was further designed to introduce a steric clash between the P119 residue and the C’ atom of T115 [15]. Remarkably, this triple mutant (T4L-L99A,G113A,R119P) inverted the population of the E and G states to 96%:4%.

Notwithstanding the success of these mutants and a number of kinetic and computational investigations into T4L [14,15,25,26,27], the actual mechanism of the G/E transition and how the three mutations modify the transition remain unclear [15]. This is mainly due to the high dimensionality of the transition itself and the lack of an automated and efficient algorithm that could dissect this G/E interconversion within affordable times and computational costs. The framework of the Markov state model (MSM) is a state-of-the-art approach to elucidate the kinetic mechanisms of complex biomolecular conformational dynamics [28,29,30,31,32,33,34,35,36,37,38,39,40,41,42,43,44,45,46]. MSMs can be constructed on a large set of short MD trajectories whose overall computational cost is lower than brute-force long simulations. The efficiency of the MSM approach could be further raised through adaptive sampling. Such speedup by adaptive sampling has been validated for small proteins with tens of residues [47,48]. However, the rational engineering of larger proteins with hundreds of residues, such as T4L, requires systematic studies of all designed mutants. Constructing MSMs for all these mutants seems far from cost-effective.

Recently, we developed an efficient path-searching method—the travelling-salesman-based automated path-searching (TAPS) method—which aims to efficiently locate high-dimensional minimum-free-energy paths (MFEPs) between two stable states of a biomolecular system at affordable cost [49]. The robustness and efficiency of this method have been demonstrated in a variety of complex systems, with sizes ranging from tens to hundreds of residues [50,51]. Though not directly providing kinetic mechanisms, the high-dimensional MFEP found by TAPS is sufficient in offering atomistic insights into the effect of a mutation on the dynamics of the protein. Its low cost also enables large-scale usage for systematic studies of all mutants involved in protein engineering.

Here, we first used TAPS to locate the MFEPs for the G/E transition of all three T4L variants. Subsequent free-energy calculations along the found MFEPs successfully revealed transition states and intermediate states that were never reported for any of the three variants. We found that, for all three mutants, the G/E transition is a three-step process composed of the flipping of the residue F114, the rearrangement of α0/α1 helices, and the final refinement of the hydrophobic cavities.

Interestingly, we found that in addition to enhancing the stability of α1, as envisioned in the original design of the double mutant, the G113A substitution also fills the hydrophobic Cavity I through the CH3 group of A113 and, therefore, reduces the free-energy barrier for flipping the residue F114. In addition, the change in the stability of the G state by the R119P mutation is minimal. Instead, the major effect of this substitution is to stabilize the E state by two sorts of new interactions: the polar contact between S117_OG_ and N132_ND2_, the salt bridge of E11_OE1_:R145_NH1_, and the polar contact between E11_OE2_ and Q105_NE2_. Meanwhile, the ring of W138 is completely flipped into the hydrophobic Cavity I, which further enhances the stability of the E state for the triple mutant. These novel insights, which were unexpected in the original mutant design, indicate the necessity of incorporating path searching into the workflow of rational protein engineering.

## 2. Methods

### 2.1. Initial Structures and Transition Paths for Three Variants of T4L

All of the molecular dynamics simulations were performed with GROMACS-2019.4 [52]. For T4L-L99A, the G state (PDB ID: 3DMV) [14] and the E state (PDB ID: 2LCB) [15] structures were resolved. For the double mutant T4L-L99A,G113A and triple mutant (T4L-L99A,G113A,R119P), their initial G/E structures were homology models based on the corresponding T4L-L99A structures as templates [53], except the E state of the triple mutant that was resolved (PDB ID: 2LC9). Then, the G state structures of the three variants were respectively solvated in a dodecahedron box with 10,331, 10,339, 10,332 TIP3P waters and 10 Na^+^/18 Cl^−^, 10 Na^+^/18 Cl^−^, and 10 Na^+^/17 Cl^−^ ions [54] (details are provided in Appendix A). The interaction within the system was described by a CHARMM36 force field [55]. After 50,000 steps of energy minimization, a 100 ps NVT simulation was performed at 300 K, followed by a 2 ns NPT equilibration to 1 atm using the Berendsen barostat [56]. For all the MD simulations, the long-range electrostatic interactions were treated by the particle-mesh Ewald method [57,58]; and the cutoff of 10 Å was used for the short-range electrostatic and van der Waals interactions. All bonds were constrained by the LINCS algorithm [59] (details are provided in Appendix A).

After that, the initial paths of the three variants for the G/E transition were generated by targeted MD (tMD) [60], with the E states as the target structures. For tMD, a total 400 ps simulation was performed for each of the three variants, and the same atom-sets for structural alignment and RMSD computation were used, with frames recorded every 0.2 ps (details are provided in Appendix A).

### 2.2. Path Optimization by TAPS

#### 2.2.1. Flowchart of TAPS Optimization

As a recently matured method, the travelling-salesman-based automated path-searching (TAPS) method can efficiently locate the minimum free-energy path (MFEP) for complex biomolecular systems [49,50]. Before performing the TAPS optimization, one needs to define the path collective variable (PCV) in a high-dimensional path [61] (see Equation (1)). The distance between any high dimensional conformation *x* and the *i-*th node of the path (*N* nodes in total), *d_x_*_,*i*_, was defined with root-mean-square deviation (RMSD), which used the same atom-sets in tMD. *λ* is a constant computed by *λ* = 2.3 (*N* − 1)/$\sum_{i=1}^{N-1}d_{i,i+1}^{2}$, where *d_i_*_,*i+*1_ is the distance between *i-*th node and its neighbor node *i* + 1.
(1)$$s=\frac{\sum_{i=1}^{N}{ie}^{-{\lambda d}_{x,i}^{2}}}{\sum_{i=1}^{N}e^{-{\lambda d}_{x,i}^{2}}},\;\;z=-\frac{1}{\lambda }\ln(\sum_{i=0}^{N}e^{-\lambda d_{x,i}^{2}})$$

Here, two components were defined with PCV [62,63], where PCV-*s* and PCV-*z* represented the position for the projection of conformation *x* on the path and the hyperplane perpendicular to the path, respectively. Because the paths other than MFEP are unstable in directions orthogonal to themselves, the node-sampling in TAPS was performed within the hyperplane perpendicular to the path through a restraint potential added on PCV-*s* via PLUMED [64]. After that, the candidate nodes for the new path were reordered via a travelling-salesman scheme (solved by the Concorde package) [65] and adjusted by the insertion and deletion of path nodes to maintain the geometric resolution of the path (more details are provided in Appendix A and Refs. [49,50]).

#### 2.2.2. Path Optimization Convergence Check

As described in our previous studies [50,51], the convergence of TAPS optimization can be conveniently checked in two ways:

(a)the distance between the path in its current iteration and the initial path, as measured by $\sqrt{z}$ [50,51,63]. See the details provided in Equation (2):

(2)$$\sqrt{z}=\sqrt{\frac{1}{N}\sum_{i=1}^{N}z_{i,\mathrm{ref}.\;\mathrm{path}}},$$
where *z_i_*_,ref. path_ denotes the PCV-z of node *i* of path α with respect to reference path;

(b)the visualization of the optimization process via the projection of all paths on a low-dimensional space generated by multidimensional scaling (MDS) [66] (more details are provided in the Appendix A and Ref. [34]).

### 2.3. Free-Energy Calculation

After convergence, we used umbrella sampling to evaluate the free-energy surface along the PCV-*s* of the found MFEP and conducted analysis by the weighted histogram analysis method (WHAM) [67,68,69]. According to the free-energy surface along the found MFEP, the transition states and intermediate states were then further identified.

## 3. Results

### 3.1. The Main Activation Mechanism for the Three T4L Variants

First, the initial paths for the G/E state interconversion were generated by tMD [60], as described in Section 2.1. We then employed TAPS to optimize the initial paths for the three variant systems. We chose all of the Cα atoms of the residues from α-helices and β-sheets for structural alignment (excluding residues 92–140, where the major conformation change occurs), and selected all of the heavy atoms of residues 92–140 for RMSD calculations (details of the parameters are provided in Appendix A). Such choices of atom-sets for structural alignment and RMSD calculations introduced few a priori assumptions about the transition mechanism.

According to the convergence analysis of TAPS by MDS [66] and $\sqrt{z}$ [50,51,63], all three T4L variant systems reached convergence to their final MFEPs (orange) within 40 to 60 iterations (Appendix A). Such convergence rates corresponded to the accumulated sampling of ~102.46 ns, ~165.48 ns, and ~72.49 ns, respectively, for T4L-L99A, T4L-L99A,G113A, and T4L-L99A,G113A,R119P (Appendix A).

Based on the MFEP located by TAPS, the free-energy landscape along PCV-*s* of the found MFEP was obtained by umbrella sampling [67,68,69]. To ensure sufficient overlap between neighbor umbrellas, extra umbrella simulations were added, with an average gap of 0.5 on PCV-s. For all three mutant systems, 2 ns of restrained sampling was performed for each umbrella (results in Appendix A). Finally, through WHAM analysis [68,69], we obtained the free-energy landscapes with the important transition states (TS) and intermediate states (IS), as shown in Figure 2A–C.

According to the free-energy landscapes, the free-energy differences between the G and E states from our simulations are very consistent with the experimental measurements: −2.80 k_B_T_TAPS_ vs. −3.51 k_B_T_Exp_ for the T4L-L99A (red), −1.38 k_B_T_TAPS_ vs. −1.94 k_B_T_Exp_ for the T4L-L99A,G113A (blue), and 2.15 k_B_T_TAPS_ vs. 3.17 k_B_T_Exp_ for the T4L-L99A,G113A,R119P (green), as shown in Figure 2D [15].

Our MFEPs also revealed that all three T4L variant systems share three major steps for their G/E transition (see Figure 2A–C). The first step was the F114 flipping (yellow) from Cavity I to Cavity II and the opening of the flipping channel between M102 and M106, as illustrated in Figure 2E. Subsequently, the rearrangement of the two α-helices (α0/α1) occurred: part of the α1 helix (residues 115 to 118) was released from α1 and merged into the α0 (Figure 2F). Finally, the hydrophobic Cvity I was refined, with F114 re-positioned. The complete set of structures for these conformational changes are shown in the Appendix A (T4L-L99A in Appendix A, the double mutant T4L-L99A,G113A in Appendix A, and the triple mutant T4L-L99A, G113A, R119P in Appendix A).

### 3.2. Enhancing the Hydrolysis of T4L through Three Substitutions

As stated in Section 3.1, the fractional populations of the G and E states for the T4L-L99A can be manipulated by introducing additional mutations [15]. However, the detailed mechanism underlying the population shift remains elusive, largely due to the lack of reliable structures corresponding to the G and E states for the T4L variants. Based on our TAPS method, the stable structures of the G and E states were identified from the free-energy landscape along the converged path optimized by TAPS (especially the G state structure of the T4L-L99A,G113A variant, as shown in Appendix A). This enables a detailed investigation into the structural differences between the G and E states for all three T4L variants (Figure 3A,B).

Although the mutants G113A and R119P were designed to change the structure and, therefore, the population of the G state, only minor differences were found for the three T4L variants (Figure 3A and Appendix A). For the T4L-L99A mutant, the hydrophobic Cavity I was fulfilled by F114 and hydrophobic residues M102/M106. In addition, a polar contact between Q105 and W138 was formed to maintain the stability of Cavity I. For the double mutant T4L-L99A,G113A, F114 retained its position, as did the residues M102 and M106. The G state structure of the double mutant T4L-L99A,G113A was very similar to that of the T4L-L99A mutant, except that the methyl group A113_CH3_ flipped into the hydrophobic Cavity I and the polar contact between Q105 and W138 was broken. The position and the orientation of W138 remained as in T4L-L99A. For the triple mutant T4L-L99A,G113A,R119P, the positions of F114, M102, and M106 remained the same, while the A113_CH3_ slightly moved away from Cavity I with the Q105-W138 contact reformed.

Nevertheless, for the E states, we have found drastic differences induced by the mutants of G113A and R119P, as shown in Figure 3B and Appendix A. For T4L-L99A, the hydrophobic Cavity I was only fulfilled with M102 when F114 flipped into Cavity II. In contrast, for T4L-L99A,G113A, the methyl group A113_CH3_ participated into the formation of the hydrophobic Cavity I, enhancing the stability of the E state. Moreover, one salt bridge (E11_OE1_:R145_NH1_) and one polar contact (E11_OE2_:Q105_NE2_) were formed with the additional R119P substitution. An additional polar contact between S117_OG_ and N132_ND2_ was found to further stabilize the hydrophobic Cavity I. Notably, the ring of W138 was completely inserted into the hydrophobic Cavity I for T4L-L99A,G113A,R119P, which is a distinct feature not observed for T4L-L99A or the double mutant T4L-L99A,G113A. Therefore, we hypothesized that these rarely reported salt bridges/polar contacts and the position of the ring of W138 are critical for the shift of the fractional population of the E state from 3% (T4L-L99A) to 96% (T4L-L99A,G113A,R119P).

To validate this hypothesis, we performed 1 μs conventional MD simulation for the G state and the E state of each T4L variant. We measured the stability of the two hydrophobic cavities (residues 92–140) by calculating their average RMSD values and the standard deviations of the RMSD, as shown in Figure 3C and Appendix A. All the RMSD values were around 2.2 Å, with a small deviation of 0.3 Å, demonstrating the stability of the G and E states. We also analyzed the percentage of the formation of the polar contact between S117_OG_ and N132_ND2_, and validated its high stability (more than 94.2%, as shown in Appendix A). In addition, the formation of the salt bridge of E11_OE1_:R145_NH1_ and the polar contact between E11_OE2_:Q105_NE2_ were also calculated. The corresponding percentages of the formation of these contacts were 83.6% and 30.8%, consistent with previous experimental results [15]. An additional 1 μs conventional MD simulation for the E state of T4L-L99A,G113A,R119P with three extra mutations (Q105L, S117A, and R145L) suggested that without these newly identified salt bridge/polar contacts, the two hydrophobic cavities (residues 92–140) become unstable, with a notably broader distribution of the RMSD value than that in the original E state of T4L-L99A,G113A,R119P (as shown in Appendix A).

### 3.3. Transition Mechanism of T4L

As illustrated in Section 3.1, the transition mechanism of the T4L from the G state to the E state can be mainly divided into three steps: the F114 flipping, the α0/α1 rearrangement, and the final refinement. To gain direct insights into the influence of the mutations on the transition mechanism, we have analyzed and discussed the details of the three main steps, respectively, as shown in Figure 4, Figure 5 and Figure 6.

#### 3.3.1. Step One: F114 Flipping

In the first step, the detailed flipping processes of F114 for the T4L-L99A (Figure 4A), the T4L-L99A,G113A (Figure 4B), and the T4L-L99A,G113A,R119P (Figure 4C) were displayed by visual molecular dynamics (VMD). For the T4L-L99A, Cavity I was destabilized by the breaking of the H-bond between W138_NE_ and Q105_OE1_. After that, the hydrophobic contact formed by the side chains of M102 and M106 was broken, simultaneously forming a flipping channel for F114, as highlighted in TS I. F114 then started to rotate toward Cavity II, coupled with an adjustment on its benzene plane to a horizontal direction, which reduced the steric effect for F114 to flip through the channel. In TS II, we also noted that the W138 slightly moved away from the M102 and reformed the H-bond with Q105. Such movement released more space for flipping F114 into Cavity II (see IS II).

Unlike the situation with T4L-L99A, similar but more feasible transition processes were found for T4L-L99A,G113A and the T4L-L99A,G113A,R119P (Figure 4B,C). With the methyl group A113_CH3_ fulfilling the empty space of Cavity I, the flipping of F114 in the T4L-L99A,G113A became easier, resulting in a lower energy barrier (~8.75 k_B_T than ~10.36 k_B_T in the T4L-L99A). For T4L-L99A,G113A,R119P, the energy barrier was slightly raised, possibly due to the stabilization of the H-bond between W138 and Q105. Such an H-bond might also restrict the rotation of A113_CH3_ into Cavity I.

#### 3.3.2. Step Two: α0/α1 Rearrangement

Following the flipping of F114, the rearrangement of the helices α0/α1 occurred (Figure 5). For the T4L-L99A, the α0 helix was rotated toward W138, reducing the size of hydrophobic Cavity I. Meanwhile, part of α1 (residues 114–117) was merged into α0 (see Figure 5A).

For the double and triple mutants, the rearrangement was completed by two steps (Figure 5B,C). At first, the 114–117 segment became partially isolated from α1 without merging with α0. Subsequently, the rotation of the α0 helix became difficult, as the hydrophobic group A113_CH3_ fulfilled the hydrophobic Cavity I. For the T4L-L99A,G113A, α0 rotated to W138 and merged with this 114–117 segment.

Unlike the situation with the single and double mutants, in the triple mutant a new stable polar contact was formed between S117_OG_ and N132_ND2_. This contact was very stable, as shown in Figure 5C. With arginine at position 119 mutated to proline, the exclusion between R119 and K83 was eliminated, resulting in a shorter distance between α1 and the helix of residues 82 to 90, as shown in Appendix A. Due to this polar contact, the rotation of α0 had to cross a higher energy barrier (~8.54 k_B_T). In addition, a salt bridge (E11_OE1_:R145_NH1_) and a polar contact (E11_OE2_:Q105_NE2_), never observed in the single or double mutant, were formed. These two interactions seemed to enhance the stability of the E state for T4L-L99A,G113A,R119P.

After the F114 flipping and the α0/α1 rearrangement, the shape of the hydrophobic Cavity I remained unstable and, therefore, required further conformational refinement.

#### 3.3.3. Step Three: Final Refinement

For the T4L-L99A, the final refinement contained a high free-energy barrier (~12.28 k_B_T), mainly involving the adjustment of the connection between α0 and α1, and the shrinkage of the hydrophobic Cavity I. Meanwhile, the ring of W138 also flipped toward solution. For the double mutant, the final refinement was merely local adjustment on the position of A113_CH3_ and the side chain of M106. The slight change of the hydrophobic Cavity I here corresponded to a lower energy barrier (~8.63 k_B_T), as shown in Figure 6B.

As shown in Figure 6C, the final refinement of the triple mutant was considerably different from that of the double mutant. With the constraint induced by the salt bridge (E11_OE1_:R145_NH1_), the polar contact (E11_OE2_:Q105_NE2_), and the highly stable polar contact between S117_OG_ and N132_ND2_, the ring of the W138 flipped inside the hydrophobic Cavity I, while the A113_CH3_ group, the M102, and the M106 almost retained their positions and orientations.

## 4. Discussion

The present case study of T4 lysozyme showed that it could be insufficient to design protein mutants based on information about a single conformational state, typically obtained via structural biology techniques. Our TAPS approach revealed not only the three-step G/E transition process of T4L (the flipping of F114, α0/α1 rearrangement, and final refinement), but also the actual interactions underlying the dominance of the E state over other states in the triple mutant, which are beyond the scope of the original design of this mutant [15].

First, the A113_CH3_ group, introduced in the double mutant T4L-L99A,G113A, reduced the energy barrier for the F114 flipping and increased the stability of the hydrophobic Cavity I in the E state, resulting in the conversion of the G:E fractional populations from 97%:3% (T4L-L99A) to 66%:34% (T4L-L99A,G113A). Second, the R119P mutation further raised the population of the E state to 96%, not through destabilizing the G state as originally believed, but by stabilizing the E state via a newly formed salt bridge (E11_OE1_:R145_NH1_) and two polar contacts (E11_OE2_:Q105_NE2_ and S117_OG_:N132_ND2_).

These results demonstrate TAPS as a promising approach in boosting rational in silico protein engineering. The overall high efficiency of automated path searching enables a rapid check of the effect of a large number of mutations on the functional dynamics of the protein, simultaneously presenting the design result and the mechanism therein.

We also noticed that previous studies indicated the existence of multiple pathways for the G/E transition of T4L. This general phenomenon in the functional dynamics of many biomolecular systems is actually calling for a sampling scheme that is capable of systematically obtaining multiple parallel MFEPs at affordable costs. Here, we used targeted MD that generated a single initial G/E path for each mutant, without searching for alternative paths. However, work in our lab continues toward a more complete sampling.

## Figures and Tables

**Figure 1 ijms-23-14628-f001:**
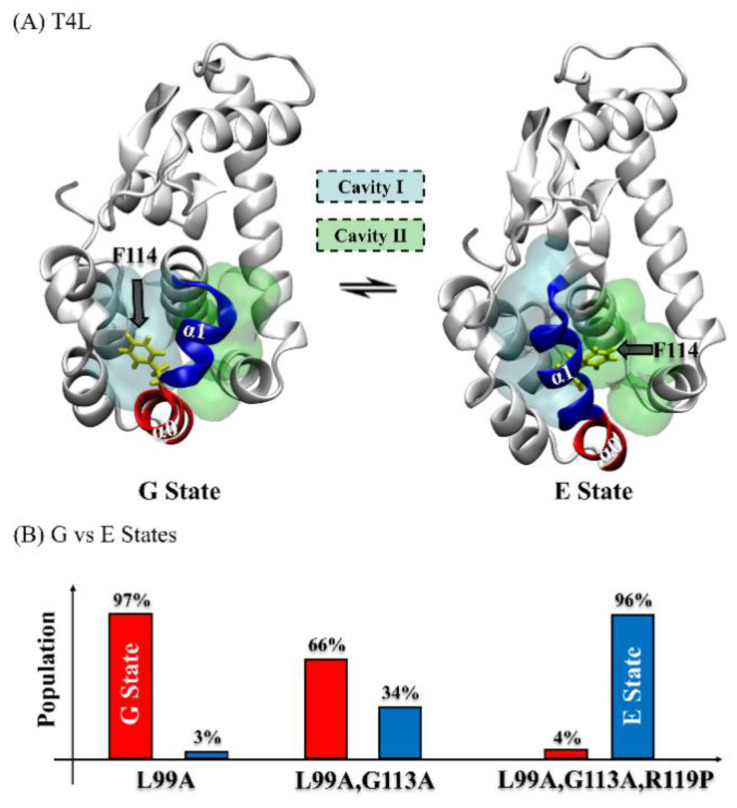
(**A**) The overall structures of the ground state (G state) [14] and the excited state (E state) [15] for T4 lysozyme, with the two hydrophobic cavities shown: Cavity I (formed by M102/M106/L133/W138, cyan) and Cavity II (formed by L84/V87/A99/V103, green). The G/E interconversion mainly involves the flipping of F114 (yellow) and the rearrangement of helices α0 (red, G state: residues 114 to 122; E state: residues 119 to 122) and α1 (blue, G state: residues 107 to 113; E state: residues 107 to 118). (**B**) The relative fractional population between the G state (red) and the E state (blue) was modulated by different mutants [15].

**Figure 2 ijms-23-14628-f002:**
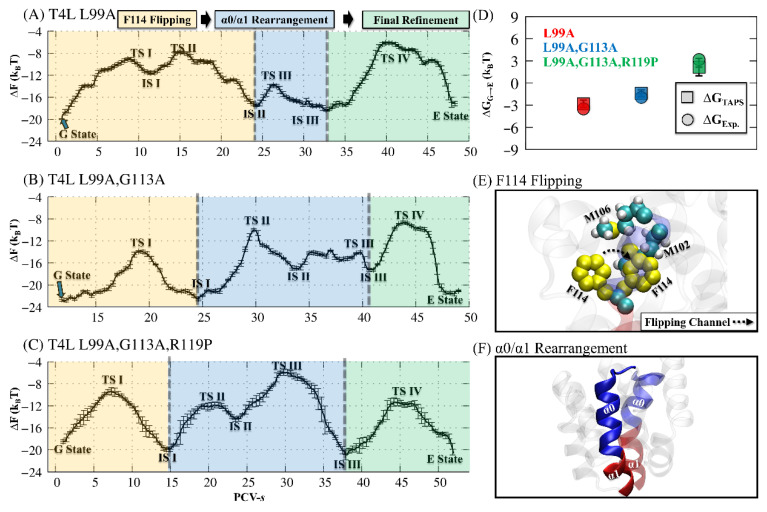
(**A**–**C**) The free energy landscapes revealed by umbrella sampling method [67,68,69] along the found MFEP for (**A**) T4L-L99A, (**B**) T4L-L99A,G113A, and (**C**) T4L-L99A,G113A,R119P. The important transition state (TS) and intermediate state (IS) are identified from the free energy landscape. The three main steps of the G/E interconversion are highlighted: F114 flipping (yellow), α0/α1 rearrangement (blue), and final refinement (green). (**D**) The free-energy differences between G state and E state (∆G_G→E_) for T4L-L99A (red), T4L-L99A,G113A (blue), and T4L-L99A,G113A,R119P (green). The squares and circles are used to label ∆G_G→E_ from the TAPS and the experiment, respectively [20]. (**E**) Illustration of flipping of F114. (**F**) Illustration of the α0(blue)/α1(red) rearrangement. All the structures are displayed by visual molecular dynamics (VMD) [70].

**Figure 3 ijms-23-14628-f003:**
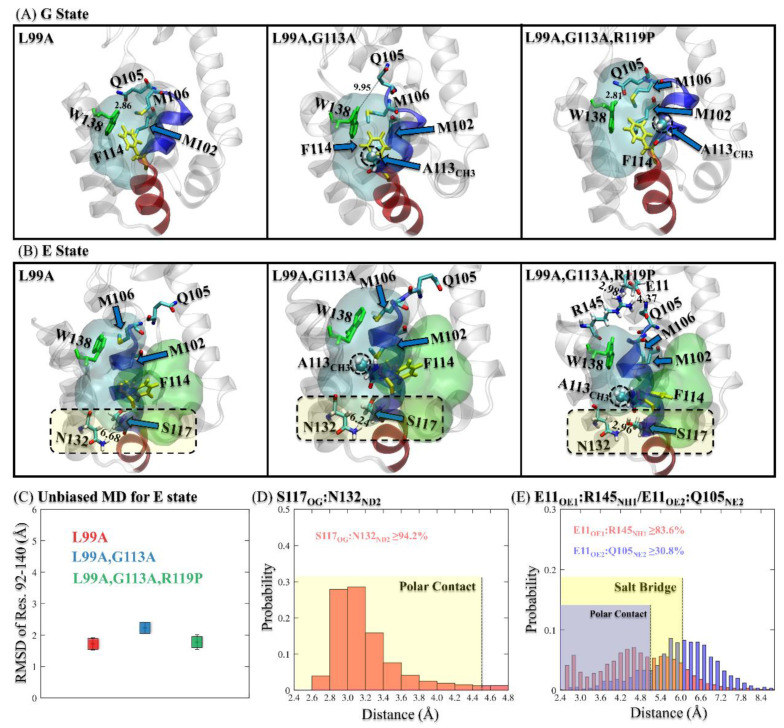
(**A**,**B**) The detailed structures of the G states (**A**) and the E states (**B**) for the T4L-L99A, the T4L-L99A,G113A, and the T4L-L99A,G113A,R119P variants, with the newly found salt bridges and H-bonds highlighted. (**C**) The average RMSDs of the heavy atoms of residues 90–140 for T4L-L99A (red square), T4L-L99A,G113A (blue square), and T4L-L99A,G113A,R119P (green square) in the 1 μs conventional MD simulations of the E states. (**D**,**E**) The probability distributions of the distances between the groups of the residues that form the polar contact S117_OG_:N132_ND2_ (**D**), the salt bridge E11_OE1_:R145_NH1_ and the polar contact E11_OE2_:Q105_NE2_ (**E**).

**Figure 4 ijms-23-14628-f004:**
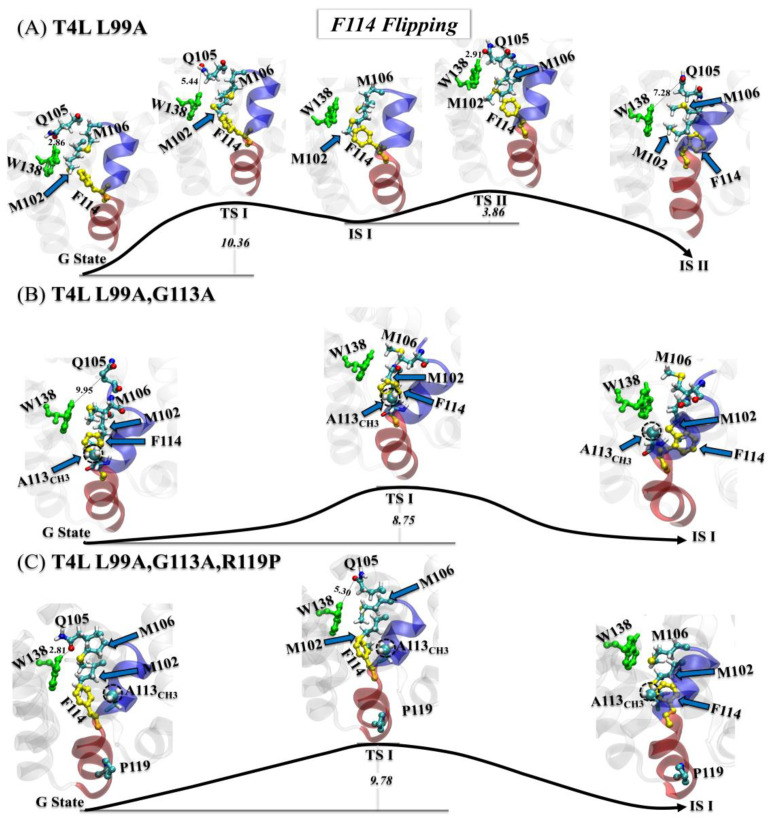
(**A**–**C**) Mechanisms of the first step (F114 flipping) of the G→E transition for T4L-L99A (**A**), T4L-L99A,G113A (**B**), and T4L-L99A,G113A,R119P (**C**). All transition/intermediate states were identified from the free-energy landscapes in Figure 2A–C, with key residues and structural segments highlighted: α0 (transparent blue), α1 (transparent red), F114 (yellow), W138 (green), and A113_CH3_ (dashed black circle).

**Figure 5 ijms-23-14628-f005:**
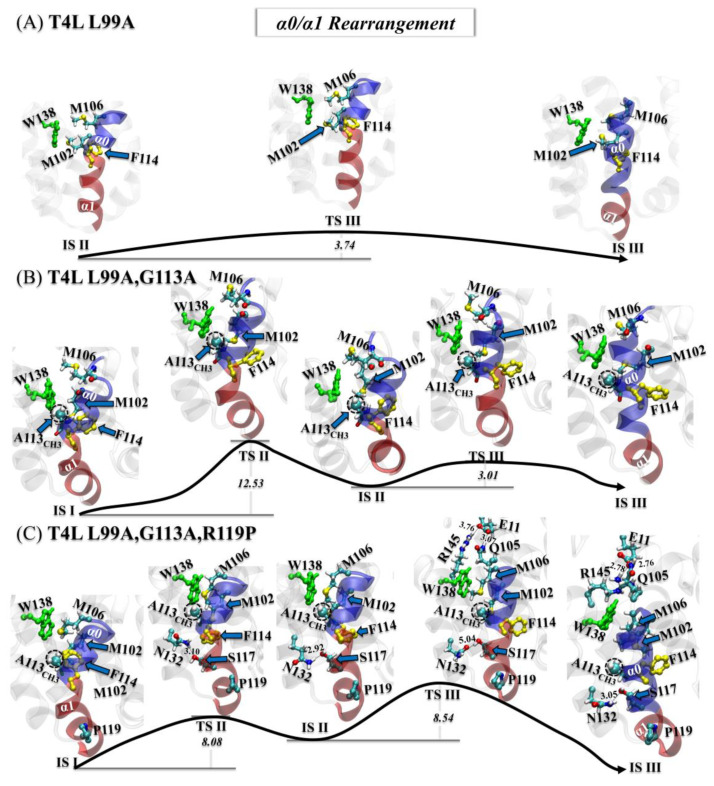
(**A**–**C**) Mechanisms of the second step (α0/α1 rearrangement) of the G→E transition for the T4L-L99A (**A**), the T4L-L99A,G113A (**B**), and the T4L-L99A,G113A,R119P (**C**). All transition/intermediate states were identified from the free-energy landscapes in Figure 2A–C, with key residues and structural segments highlighted: α0 (transparent blue), α1 (transparent red), F114 (yellow), W138 (green), and A113_CH3_ (dashed black circle).

**Figure 6 ijms-23-14628-f006:**
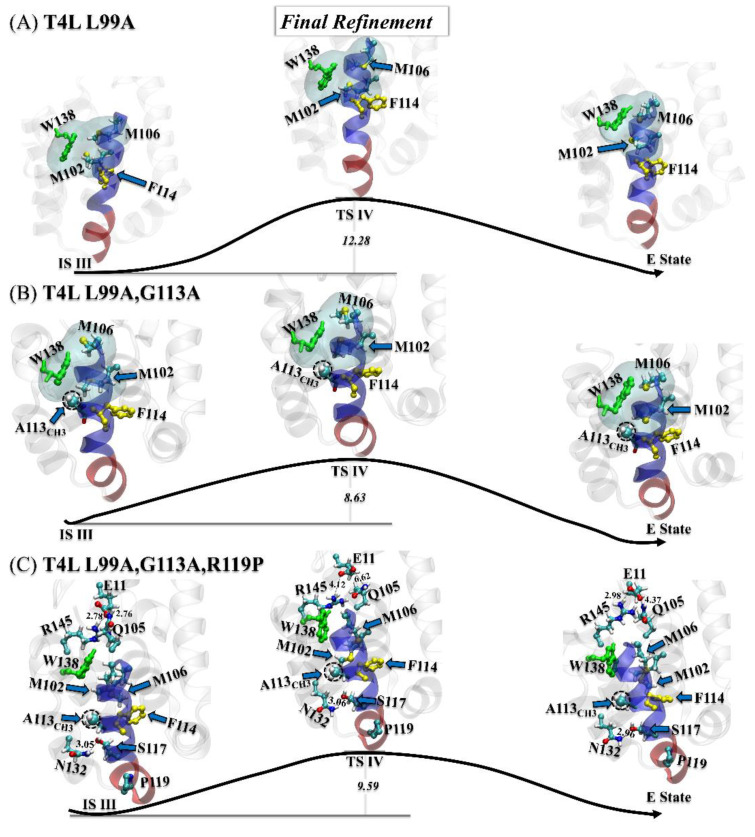
(**A**–**C**) Transition mechanisms of the third step (final refinement) of the G→E conversion for the T4L-L99A (**A**), the T4L-L99A,G113A (**B**), and the T4L-L99A,G113A,R119P (**C**). All transition/intermediate states were identified from the free-energy landscapes in Figure 2A–C, with key residues and structural segments highlighted: α0 (transparent blue), α1 (transparent red), F114 (yellow), W138 (green), and A113_CH3_ (dashed black circle).

## Data Availability

The data presented in this study are available in this article or its Appendix A.

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
