# Peer review of "Automated Path Searching Reveals the Mechanism of Hydrolysis Enhancement by T4 Lysozyme Mutants"

_ijms, 2022, doi:10.3390/ijms232314628_

Round 1

Reviewer 1 Report

In this report, the authors investigated mechanism of hydrolysis enhancement by T4 lysozyme mutants (L99A, L99A/G113A, L99A/G113A/R119P) using in silico approach, i.e., Travelling-salesman based Automated Path Searching (TAPS) developed by the authors. These computational data revealed a transition mechanism from ground to excited states via three-step processes. Especially, R119P mutation increases the population of E-state to 96% through its stabilization via newly formed polar interaction including E11-R145, E11-Q105, S117-N132, not through destabilizing the G-state as originally believed. The presented results would provide valuable insights in the field of protein engineering. This reviewer would recommend this manuscript for publication in International Journal of Molecular Sciences. However, this reviewer has several concerns that could be addressed in revision of the manuscript.

Minor points:

Lines 8-24: The author should not use abbreviation, i.e., E and G states, in the abstract.

Line 9: Sentence “Due to~ regard as a” is incomplete. 

Line 50: Alanine should be small capital. 

Major points:

In this manuscript, the authors proposed a new transition mechanism whereby R119P mutation facilitates the stabilization of E-state via salt bridge and polar interaction mediated by E11-R145, E11-Q105, S117-N132. To demonstrate this hypothesis, further mutational analysis of these residues will be necessary using theoretical and or experimental approach(es). 

Lines 85-87: Unlike Arg (R145), Gln (Q105) does not make a salt bridge with Glu (E11). This should call as hydrogen bond or polar contact. The authors should revise the expression throughout the manuscript. 

Reviewer 2 Report

T4 Lysozyme has been a representative model system for studying protein dynamics and functions. Scientists have created mutations to shift the structure of T4L from its ground state to its excited state. Despite extensive studies out there studying the three mutations, the actual mechanisms of the G/E transitions in these mutants remained unclear. In this manuscript, the authors used their previously described TAPS method to reveal the molecular mechanisms. The authors found a three-step process involving the flipping of F114, alpha0/alpha1 rearrangement, and final relaxation. 

The manuscript was well written and well-formatted. The results were clearly explained and well-presented. I suggested that the manuscript accepted in its present form.

Reviewer 3 Report

In this study, the authors probe the effects of mutations (L99A,G113A,R119P) on the Bacteriophage T4 lysozyme (T4L), regarding its mechanism of action and catalytic activity. This study should be of interest to a broad research community, as it firstly introduces the in-house built method of "path searching" TAPS. Secondly, It reports on the dynamics of T4 Lysozyme, which is a key antimicrobial agent in the food industry, but also a model enzyme for structure-function relationship. There are some points however that need to be addressed prior to the consideration for publication:

(1) I do not agree that there is a high demand of computational resources for constructing MSMs. In fact, many short MD trajectories can be used as input for the MSM models, that can be validated. An adaptive sampling algorithm can be built for the efficient and low-computational cost sampling of a protein configurational space (J. Chem. Theory Comput. 2020, 16, 12, 7915–7925; J. Chem. Theory Comput. 2010, 6, 3, 787–794).

(2) The GROMACS version employed should be reported

(3) The authors have used targeted MD simulations, and umbrella sampling for the configurational space sampling of T4. Thus, a predefined pathway of threes steps (mechanism of action) is enforced. How do the authors avoid bias in their approach? This is especially important in the case of mutations with a questionable starting structure computationally predicted and not experimentally resolved. This should be discussed.

(4) The authors report that "Although the mutants G113A and R119P were designed to change the structure and therefore the population of the G state, only little differences were found for the three T4L variants". How confident are the authors that their predicted structures are adequately minimized, equilibrated, or can viewed as ensemble averages of the respective states G/ E? Structure prediction based on mutations is not a trivial task, as an extensive sampling might be needed and the starting structure for the TAPS method could have introduced a bias in the result.

Based on the above, I recommend a revision of the study/ manuscript.

Round 2

Reviewer 1 Report

In the revised manuscript, the authors performed an additional long-time MD simulation, supporting their hypothesis regarding a new transition mechanism mediated by R119P. They adequately responded to this reviewer's query.  

Author Response

This reviewer suggest that we have successfully addressed the previous issue of validating our found mechanism but need to perform additional check of gramma and spelling. We sincerely thank the help of this reviewer and have carefully revised the manuscript to correct all language problems.

Reviewer 3 Report

The authors have made amendments to the manuscript. I believe that a clear sentence stating that "A predefined pathway of threes steps is enforced for the configurational space sampling of T4. No other alternative pathways were probed, or inter-compared, so other T4 configurations might exist. However, work in our lab continues towards a more concise sampling"
